# Biomimetic Self-Assembled Chiral Inorganic Nanomaterials: A New Strategy for Solving Medical Problems

**DOI:** 10.3390/biomimetics7040165

**Published:** 2022-10-14

**Authors:** Rong Wei, Xueying Gao, Ziwei Cao, Jing Wang, Yu Ma

**Affiliations:** 1School/Hospital of Stomatology, Lanzhou University, Lanzhou 730000, China; 2Key Laboratory of Dental Maxillofacial Reconstruction and Biological Intelligence Manufacturing, Lanzhou University, Lanzhou 730000, China

**Keywords:** inorganic, chirality, biomimetic, self-assembly, biochemical application

## Abstract

The rapid expansion of the study of chiral inorganic structures has led to the extension of the functional boundaries of inorganic materials. Nature-inspired self-assembled chiral inorganic structures exhibit diverse morphologies due to their high assembly efficiency and controlled assembly process, and they exhibit superior inherent properties such as mechanical properties, chiral optical activity, and chiral fluorescence. Although chiral self-assembled inorganic structures are becoming more mature in chiral catalysis and chiral optical regulation, biomedical research is still in its infancy. In this paper, various forms of chiral self-assembled inorganic structures are summarized, which provides a structural starting point for various applications of chiral self-assembly inorganic structures in biomedical fields. Based on the few existing research statuses and mechanism discussions on the chiral self-assembled materials-mediated regulation of cell behavior, molecular probes, and tumor therapy, this paper provides guidance for future chiral self-assembled structures to solve the same or similar medical problems. In the field of chiral photonics, chiral self-assembled structures exhibit a chirality-induced selection effect, while selectivity is exhibited by chiral isomers in the medical field. It is worth considering whether there is some correspondence or juxtaposition between these phenomena. Future chiral self-assembled structures in medicine will focus on the precise treatment of tumors, induction of soft and hard tissue regeneration, explanation of the biochemical mechanisms and processes of its medical effects, and improvement of related theories.

## 1. Introduction

Chirality is an intrinsic property of the natural world, ranging from that which characterizes the DNA double helix at the nanoscale to that which characterizes the macroscopic spiral arms of galaxies [1]. In the past two decades, our understanding of chirality has rapidly evolved from encompassing only simple chiral organic compounds to encompassing the spontaneous organization of metal or ceramic nanoparticles induced by chiral organic compounds as templates [2]. The wide array of research systems and dimensions has greatly expanded the functional categories of inorganic particles [3]. Excluding the fields of chiral optics and chiral catalysis, the functionality of chiral structures has been fully exploited and utilized to replace applications of traditional materials. In the medical field, the influence of chiral assemblies is gradually increasing [4,5]. Crucial trace metals in chiral assemblies show unusual special properties that contribute to their role in physiological processes [6,7]. There exists the need to define the mechanisms of action of these chiral structures and the metals themselves that are present on both cells and microscopic matrix microstructures. Characterizing these mechanisms will require determining both the physiological activity of the metal itself and the induced selection effect of chiral polarization [8,9]. It should be noted that the self-assembly process is crucial for the formation and regulation of chiral structures, and the use of a system of electrostatic force or other non-covalent bonding forces to achieve spontaneous organization from the nanoscale to the microscale is a typical biomimetic process [10,11]. The process is equivalent to the self-planning and organization of the formation of microstructures that is inherent in all physiological activities in nature [12,13,14,15]. Self-assembled nanostructures in nature often have a mechanical efficiency exceeding 90%, whereas self-assembly by artificial intervention, while less efficient, can be harnessed to better control the self-assembled structures and their accuracy, while their corresponding energy efficiency is also much higher than that of non-self-assembled processes [16]. The reported diversity of chiral self-assembly structures—polyhedron [17], tube [18], petal [19], flower [20], helix [21], rod [22] and hedgehog chiral particles [23]—with multiple levels of a complex structure (Figure 1a–c), is the basis for their significant functional advantages across multiple domains. Such advantages capitalize on the effects of chiral self-assembly, which impact basic cellular behaviors, such as cell adhesion and proliferation, the differentiation-induction of stem cells, and the regulation of tumor cell fates [24,25]. Various forms of chiral self-assembly, including simple structures and the complex assembly of topologies, have shown surprising potential in the field of biomedical science [26]. Here, we review these chiral self-assembled structures at different scales and morphologies as obtained by diversified synthetic methods, and we outline their distinctive synthetic approaches. We focus on the application of chiral self-assembled structures in the medical field, especially the influence of chiral self-assembled structures on the basic behaviors of cells, including cell adhesion and proliferation, the absorption of surface proteins, and osteogenic differentiation. Meanwhile, we also summarize the progress in their applications for molecular probes, such as enantiomeric amino acid and reactive oxygen species (ROS) recognition and tumor therapies through inducing the autophagy and apoptosis of tumor cells and the photothermal effect [27,28]. This review also looks forward to the existing boundaries limiting their future applications and the challenges posed by using chiral self-assembled structures in the biomedical science field.

## 2. Synthesis of Biomimetic Self-Assembled Chiral Structures

Organic small molecules, inorganic particles, and nanoparticles can form complex and orderly structures through self-assembly. Inspired by this micromechanical behavior in nature, artificially inducing the spontaneous organization of microcomponents by regulating the interaction of particles in chemical or physical reactions represents an effective synthesis method to mimic nature’s self-assembling behavior [29,30]. Recent research on chiral self-assembly has focused on inorganic nanoparticles and their assembly bodies, whereby chiral building blocks such as DNA, amino acids, lipids, and sugars play an important role in inducing metal ion assembly [31,32,33]. Typical chiral self-assembly structures are ubiquitous from the nanoscale to the micron scale. While typical nanoparticles have symmetrical structures (such as spheres and cubes), asymmetric defects, vertices, and voids cause nanoparticles to exhibit chiral geometry, which is the chiral nature of nanoparticles. For example, screw dislocation-mediated growth is responsible for the chiral polyhedral shape formation of tellurium nanocrystals grown from solution [34]. Further, it was found that two Au nanorods can form strongly chiral nanoscale systems. Their chiral properties originate from the small dihedral angle between two adjacent nanorods, which breaks the centrosymetric nature of the two parallel identical cylinders [35]. The second type of chiral nanostructure is the induced chirality of chiral ligand-modified inorganic nanoparticles. There have been many studies about the chiral synthesis related to various nanostructures, especially nanorods. For example, the cholic acid-induced anisotropic epitaxial growth of chiral CdSe/CdS nanorods (CCCNs) and their self-assembly into chiral nematic-like films (CNFs) via an evaporation-induced self-assembly route [36] has been reported. Discrete gold nanorods (Au NRs) with strong chiroptical responses in the visible and near-infrared region were synthesized by a seed-mediated approach in the presence of L- or D-cysteine by Zheng et al. [37]. In addition, nanoclusters formed by the specific arrangement of nanoparticles can also form chiral structures. Previous research has reported the 2D self-assembly of ligand-capped Au15 nanoclusters into mono-, few-, and multi-layered sheets in colloidal solution. This 2D self-assembly is caused by the 1D dipolar attraction common in nanometer-sized objects and produces an asymmetric van der Waals attraction [38]. Most existing chiral nanoparticles are formed from chiral transfer reactions in which the metal nanoparticles are modified by chiral ligands. Common methods include the reduction of metal salts and the replacement of metals by chiral complexes. For example, undecagold cluster compounds were synthesized by the chemical reduction of the corresponding precursor complex, Au2X2 (BINAP) [39]. Additionally, ligands such as L- or D-aspartic acid (Asp), L- or D-proline (Pro), and cysteine are used for the transmission of chiral signals [40,41]. 

Besides, DNA origami is another tactic used to induce chirality, taking advantage of its programmability, flexibility, and high assembly efficiency. This strategy mainly involves binding DNA to AuNPs through the prethiolation of strands [42] or combining it with other metal via an electrostatic force [43]. Researchers have found that self-assembly can be controlled through the integration of gold nanoparticles [44]. Similarly, Kuzyk et al. used DNA origami technology to synthesize a gold nanoparticle helical structure [45]. By designing the ‘X’ pattern of the arrangement of DNA capturing strands (15 nt) on both sides of a two-dimensional DNA origami template, AuNRs functionalized with the complementary DNA sequences were positioned on the origami and were assembled into AuNR helices, with the origami intercalated between neighboring AuNRs (Figure 2) [46]. Other structures were also reported, including a DNA pyramid containing four AuNPs linked with DNA stands that can produce a stronger CD response compared to chiral metal clusters or NPs [47]. DNA-bound NPs offer another platform to create new conjunction sites. Lan et al. successfully combined K21 cyanine dyes with DNA-assembled AuNPs, which also presented a stronger CD signal [48].

It is also feasible to obtain chiral metal nanoparticles by treating chiral organometallic precursors with light, sound, and magnetism. Srivastava et al. produced cadmium telluride (CdTe) nanoparticles to form both left and right chiral helical structures using illumination with visible light, while Xu et al. irradiated precursor particles with circularly polarized light (CPL) and linearly polarized light [49,50]. Liquid exfoliation by ultrasonic treatment and modulation by magnetic fields have also been used to treat chiral organometallic precursors [51,52].

The basic chiral components in chiral structures beyond the nanoscale can accurately transmit asymmetric polarization into topologies due to the action of structural guides, and the corresponding isomeric self-assemblers can accurately form structures with mirror symmetry under the same synthesis conditions. Unlike chiral nanoparticle formation, the formation of chiral topological self-assemblies involves not only the expansion of ligand-directed chiral metal nanoparticles, but also the amplification of chiral signals [53,54]. This must capture the balance between the self-sealing end of the structure and its self-limiting point [55]. The differentiated structures of chiral self-assemblies are characterized by their chiral ligands, solvents, composition, synthesis temperature, and growth time [56,57,58]. Both left-handed and right-handed chiral cysteines can self-assemble with stable CdTe to produce chiral mesoscale helices. These helical structures contain multi-level features ranging from the nanometer scale to the micrometer scale, and the termination of the length and diameter of the helix demonstrates the self-limiting nature of the structure, which is related to the directional attachment of inorganic nanoparticles [59]. On this basis, Jiao et al. obtained a chiral helix structure through the self-assembly of semiconductor nanoparticles and then studied the resulting structural changes of the chiral helix by adjusting the solvent composition, pH, and ligand density. The modulation of the coordination and hydrogen bonds enabled controllable chirality deviation and greatly improved assembly yields [60]. The efficient assembly of chiral penicillamine and ZnS nanoparticles may have formed primary chiral supraparticles, which were then further co-assembled by glutathione-modified gold nanoparticles and ZnS supraparticles, thereby finally forming two types of composite assemblies of ZnS-Au supraparticles [61]. Jiang et al. created a hierarchically organized chiral biomineral structure of calcium carbonate whose chirality could be switched by a single L-enantiomer of an amino acid, resulting in a chiral biomineral structure that closely resembled the pathological inner ear stone [62]. Ribbons of stacked, board-shaped cadmium selenide (CdSe) nanoplatelets (NPLs) were twisted upon the addition of an oleic acid ligand, leading to chiral ribbons that reached several micrometers in length and displayed a well-defined pitch of ~400 nm [63]. Singh et al. found that under carefully controlled conditions, cubic nanocrystals of magnetite self-assembled into arrays of helical superstructures in a template-free manner. Depending on the density of the nanocrystals, the researchers identified different types of self-assembled superstructures, including one-dimensional belts and single, double, and triple helices [64]. More complex chiral self-assembled structures known as hedgehog-like chiral superstructures were also successfully synthesized and the perfect symmetry of the enantiomers was maintained at the micrometer scale [65]. The core mechanism underlying both the structure formation and stabilization stems from the electrostatic repulsion and elastic confinement. In addition, Kotov and his colleagues developed graph-theoretic methods to assess the complexity of chiral self-assemblies and provided guidelines for evaluating the intrinsic correlation between the optical asymmetry factors of chiral structures and their particle complexity [66]. In conclusion, the abundance of the synthesis methods used to produce chiral self-assembled structures has paved the way for the synthesis of diverse chiral structures and their subsequent applications in biomedical fields.

## 3. Medical Applications of Biomimetic Chiral Self-Assembled Particles

Most of the physiological functions of living organisms are highly dependent on chiral assemblies (such as DNA and RNA) rather than on monochiral biomolecules (such as nucleotides) [67,68,69]. Increasingly more studies have shown that films and nanoparticles modified with enantiomers play important regulatory roles in biological processes, such as cell adhesion, differentiation, protein and DNA adsorption, cytotoxicity, and genotoxicity, among other biological processes [70,71,72].

### 3.1. Effects on Cell Adhesion, Proliferation, and Differentiation

Numerous studies have shown that cells can interact with chiral surface molecules and exhibit different effects on enantiomers. Zhao et al. designed monolayer chiral Au nanoparticle (NP) films modified with L- and D-penicillamine (Pen), respectively (Figure 3a) [73]. The L-Pen-NP films could promote cell adhesion and accelerate cell proliferation compared with the D-Pen-NP films. Meanwhile, there was a greater degree of differentiation of the cells grown on the surface of the L-Pen-NP films. L/D PEFG (phenylalanine-based gelators) self-assemble to form cross-scale nanofibrous hydrogels with enantiomers of molecular chirality and supramolecular chirality [25]. Through studying the effects on the behaviors of three different cells, Dou found that the shift away from an unordered to ordered assembly and the increase in roughness with left-handed chiral aggregates led to increased cell proliferation, while right-handed chiral aggregates caused the opposite outcome, which indicates that left-handed self-assemblies play a key role in cell adhesion and proliferation. Through the bottom-up process of self-assembly, molecular chirality can be amplified to supramolecular chirality, thereby regulating cell behaviors with finer control, and this control is thought to be coordinated by the adsorption of proteins on the surface of the material. In addition to its effects on cell adhesion, proliferation, and protein adsorption, the L-chiral self-assembled structure with molecular chirality, as well as nanofibers with supramolecular chirality, can promote the expression of osteogenic proteins on the surfaces of dental pulp stem cells, which shows a positive effect on their differentiation. However, Dong et al. found that asymmetry in the rotation direction of the cytoskeleton caused by external chiral geometry can cause different levels of contractility in human mesenchymal stem cells, especially the right-handed geometry, which performs significant roles in cell proliferation, stem cell maintenance, differentiation, and migration [74]. Two different shapes of AuNPs, i.e., gold nanocubes (AuNCs) and gold nanooctahedras (AuNOs), were synthesized using L/D-valine as the chiral center (Figure 3b) [75]. Corresponding studies have shown that both the shape and chirality of nanoparticles can affect cellular uptake and migration, and cell diffusion is also chirality-associated, while cytotoxicity was found to be mainly affected by the amounts of ingested nanoparticles and was positively correlated with ROS (reactive oxygen species) levels. 

### 3.2. Molecular Probes

Enantiomerically pure molecules play an important role in medicine [76,77], pharmacology [78,79], and chemical engineering [80,81,82,83]. In this context, the identification, detection, and separation of chiral compounds have long been the subjects of research. Previously, the chiral identification of enantiomers mainly relied on chromatographic separation [84,85,86], but the detection conditions involved were harsh and the process was time-consuming [87]. Fortunately, enantioselective fluorescent identification is a new method for chiral identification that features high sensitivity and a simple operation [88,89]. At present, a series of fluorescent probes composed of nanomaterials and supramolecules are widely used in chiral recognition. Valmik et al. synthesized a novel solid luminescent material with high thermal stability, photostability, and color-tunable properties by fusing the 2-benzoylfuran component with an aromatic unit with solid-state luminescence properties [90]. One of the forms features many strengths: it had good biocompatibility, a high fluorescence quantum yield, a large Stokes shift, was found to be well-internalized and uniformly dispersed in the cytoplasm of MDA-MB-231 cancer cells, showed high fluorescence intensity, and had potential as a chiral fluorescent probe. Jiang et al. proposed an electrochemical molecular probe based on penicillamine-modified small gold nanoparticles for the chiral recognition of 3,4-dihydroxyphenylalanine, and such enantioselectivity can be improved by adjusting the size of the penicillamine-gold nanoparticles [91]. Nanocones formed by Au-Cu9S5 nanoparticles, Ag2S, and upconverting nanoparticles can be used for the ultrasensitive quantitative detection of microRNAs in living cells, as well as their corresponding bioimaging in vivo [92]. Du et al. synthesized a BINOL-based chiral aldehyde containing hydrophilic polyethyleneglycol (PEG) groups, which can be used as a fluorescent probe for amino acids in aqueous solutions due to its highly enantioselective fluorescence effect on various amino acids after being combined with zinc ions [93]. 

Reactive oxygen species (ROS) are continuously produced by the various metabolic activities of the body and play a vital role as important signaling molecules at low concentrations [94,95]. However, the oxidation of biomolecules like proteins, lipids, and nucleic acids can severely damage cells [96,97], and high levels of ROS may cause the loss of mitochondrial cardiolipin and lipid peroxidation [98], inducing autophagy. By regulating beclin1 and Atg7, two proteins known to regulate autophagy, H_2_O_2_ can mediate autophagy [99,100]. Another pathway that induces autophagy is the activation of AMPK, which inhibits the activity of mTOR [101,102,103,104]. Chen et al. synthesized L/D-Co(OH)_2_ NPs and modified them with fluorescent molecules (Alexa Fluor 568 (AF568)) to achieve the quantitative detection of ROS levels in vivo under the dual signal of CD and MRI [105]. The study found that the D-Co(OH)_2_ NPs performed with a lower cytotoxicity and higher cellular uptake compared to the L-Co(OH)_2_ NPs, and because of the denser distribution of the D-Co(OH)_2_ NPs in the cellular microenvironment, they also had a stronger ROS detection ability at low concentrations.

### 3.3. Tumor Treatment

Many self-assembled chiral structures are used in anti-tumor therapies, whereby their primary mechanisms of action are through inducing the autophagy and apoptosis of tumor cells. Some anti-tumor effects can also be achieved through the photothermal effect. L/D-cysteine-modified Cu2-xS nanocrystals (prepared by sacrificial template–ligand exchange) can induce the production of a large number of reactive oxygen species in tumor cells, thereby inducing autophagy to achieve anti-tumor effects, and autophagy induced by D-Cu2-xS, in particular, is extensive [106]. This nanostructure may also have a killing effect on tumor cells via photothermal action. By preparing single crystals of chiral tetranucleate copper(II)-based complexes (TNCu-A), Hou et al. found that these could induce apoptosis in human triple-negative breast cancer MDA-MB-231 cells in a mitochondria-dependent manner in vitro [107]. This effect was achieved by inhibiting the anti-apoptotic protein Bcl-2 while upregulating the expression of the pro-apoptotic proteins caspase-9 and Bax. Furthermore, these could inhibit tumor angiogenesis, thereby inhibiting tumor development and progression. Chiral folic acid (FA)-conjugated CdTe/CdS quantum dots could be specifically recognized and taken up by breast cancer cells to subsequently induce the release of the apoptotic factor caspase-9 into the cytoplasm through mitochondrial membrane potential depolarization, and then trigger a cascade reaction to activate caspase-3, which further promotes cellular apoptosis (Figure 3c) [108]. In addition, FA-Cys-CdTe/CdS QDs can selectively induce cancer cell apoptosis by disrupting the α-helical structures in cancer cell proteins. In addition to killing tumors via the mechanism involving chiral self-assembly and the chiral effect, the photothermal effect has been widely reported to kill tumor cells and inhibit tumor cell metastasis based on noble metal-based nanocomposites. The photothermal effect of chiral self-assembled structures comprises an active area of research. Miao et al. synthesized a chiral molecule-induced molybdenum oxide nanoparticle, applied its photothermal properties, and found it to act as a chirality-dependent visible and near-infrared (NIR) photosensitizer for HeLa cell photothermal therapy (PTT) in an in vitro setting [109]. Circularly polarized NIR light radiation also exerts a significant effect on tumor ablation in vivo, and this tumor ablation effect may be closely related to tumor cell apoptosis. These results provide relevant strategies for the application of chiral optics and chiral optogenetics in clinical medicine. Chiral molybdenum (Cys-MoO3-x) nanoparticles (NPs) can be actively taken up by oral squamous cell carcinoma cells and show great cytotoxicity on cell viability under 808 nm laser irradiation [110]. In addition to inducing autophagy and the apoptosis of tumor cells via the photothermal effect as part of tumor therapies, there exist some anti-tumor effects of chiral self-assembled structures that focus on immune-induced responses. For example, Wang et al. obtained prismatic cube-shaped chiral nanoparticles by adding chiral molecules to gold nanoparticles and found that L-NPs could significantly inhibit tumor growth in C57BL6 mice and improve their survival rate [111]. The study also found that the mechanism underlying this phenomenon was that the L-NPs stimulated innate immunity by activating natural killer (NK) and CD8 + T cells and enhanced acquired immunity. In addition to the above effects, the nanoparticles, especially the L-type, also enhanced the cytotoxicity of NK cells and promoted the apoptosis of tumor cells to achieve their anti-tumor therapeutic effects.

## 4. Conclusions and Outlook

The array of methods used for the synthesis of multidimensional chiral self-assembled structures and the corresponding research interest for their applications in the biomedical field has grown dramatically [112]. We have reviewed their synthetic pathways including chiral self-assembled nanostructures and superstructures, as well as important advances that have been made in the production of chiral self-assembled structures for the control of cell behavior, fluorescent probes, and tumor therapies. Studies have found that the application of chiral self-assembled structures in medicine primarily depends on chiral selectivity functioning to regulate biological processes. However, many challenges remain to be solved, such as the lack of clarity that still exists regarding the formation mechanism employed by topological chiral self-assembly isomers, as well as the rules and amplification mechanisms underlying chiral signal transmission. In the existing studies, the biological effects of self-assembled structures of different systems (such as the results of tumor treatment) are quite different, and there are certain conflicts in their mechanisms. For example, in some studies, left-handed chiral self-assemblies can effectively inhibit tumors, which is attributed to the induction of autophagy; conversely, one study found a more substantial tumor-killing effect of the right-handed self-assembled structure. It should be noted that some recent chiral photonics studies have presented results indicating the existence of a chirality-induced selection effect of chiral structures on photons [113,114]. This poses the following question: is this effect related to the core mechanism of the underlying phenomenon applied in the biomedical field? In addition, research is currently lacking on the biological effects of chiral structures at larger scales, such as micron-scale chiral self-assembled structures. The in vivo toxicity, immunogenicity, and metabolic pathways of inorganic nanoparticles, especially for chiral self-assembled structures containing chromium and gold, will restrict their clinical applications. To enrich the basic concepts and theoretical models regarding the mechanisms of chiral self-assembly, to clarify the precise biological effects and molecular mechanisms employed by chiral self-assembly structures, and to seek new biologically safe and stable chiral self-assembly structures in vivo will comprise the focus and primary direction of future research.

## Figures and Tables

**Figure 1 biomimetics-07-00165-f001:**
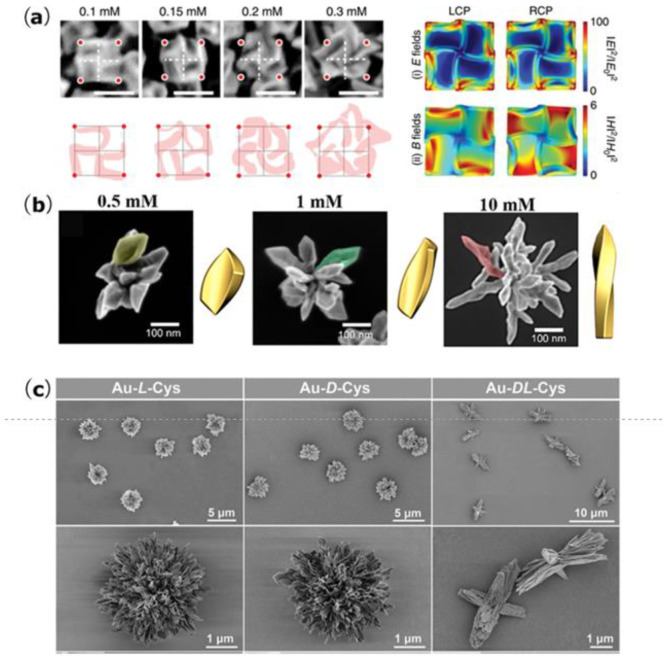
(**a**) SEM images of a rhombic dodecahedral shape under different cysteine concentrations ranging from 0.1 to 0.3 mM (reproduced with copyright permission from [17], Springer Nature). (**b**) Synthesis and morphology of the outspread petal-like structure (reproduced with copyright permission from [19], American Chemical Society). (**c**) SEM images of Au–L-Cys and Au–D-Cys coccolith-like particles and Au–DL-Cys kayak particles with low magnification, and enlarged SEM images of Au–L-Cys, Au–D-Cys, and Au–DL-Cys kayak particles (reproduced with copyright permission from [23], The American Association for the Advancement of Science).

**Figure 2 biomimetics-07-00165-f002:**
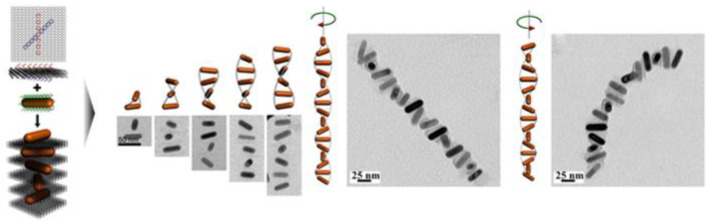
Au nanorod (AuNR) helical superstructures (helices) with tailored chirality (reproduced with copyright permission from [46], American Chemical Society).

**Figure 3 biomimetics-07-00165-f003:**
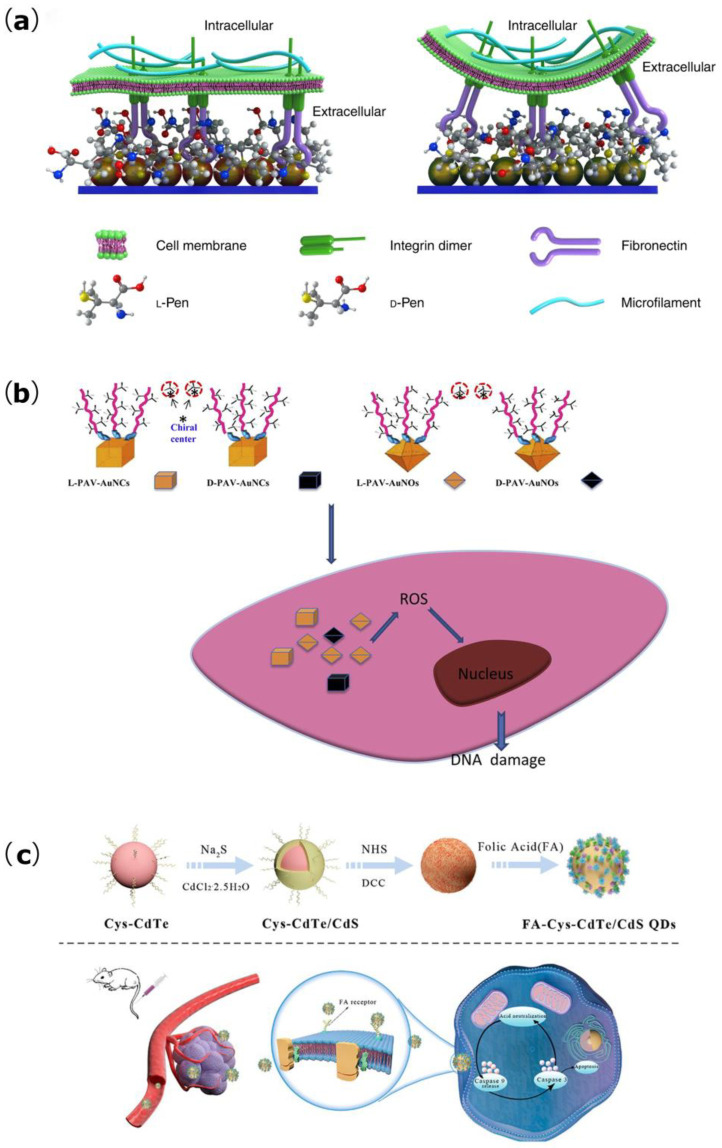
(**a**) Schematic presentation of the regulation of NG108-15 cell adhesion by the stereospecific interaction between fibronectin and the chiral NP film (reproduced with copyright permission from [73], Springer Nature). (**b**) The structure and surface chirality at the nanoscale can influence cytotoxicity and genotoxicity (reproduced with copyright permission from [75], Elsevier Ltd.). (**c**) Synthesis of chiral FA-Cys-CdTe/CdS QDs and illustration of their effect in synergistic cancer therapy by the caspase-dependent apoptosis pathway (reproduced with copyright permission from [108], American Chemical Society).

## Data Availability

Data sharing is not applicable to this article as no new data were created or analyzed in this study.

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
