# Peer review of "Biomimetic Self-Assembled Chiral Inorganic Nanomaterials: A New Strategy for Solving Medical Problems"

_biomimetics, 2022, doi:10.3390/biomimetics7040165_

Round 1

Reviewer 1 Report

In this review intitled “Biomimetic self-assembled chiral inorganic structures: a new strategy for solving medical problems” authors report various forms of chiral self-assembled inorganic structures and applications in biomedical fields of biomimetic chiral self-assembled particles. It is a valuable source of information in this field. Only technical correction should be done:

Line 31-32 “chiral organics” have to be corrected in “chiral organic compounds”

Line 92-93 “l- or d-“ is “L- or D-“

Line220 “H2O2” is “H2O2

Line 225 and 226: “d-Co(OH)2 NPs”  have to be corrected in: “D-Co(OH)2 NPs”

Line 226:” l-Co(OH)2 NPs” have to be corrected in: “L- Co(OH)2 NPs”

Figure 1 and 2 are never cited in the text

Line 302-303: “The in vivo toxicity, immunogenicity, and  metabolic pathways of inorganic nanoparticles; especially for chiral self-assembled structures containing chromium and gold; will restrict their clinical applications” have to be corrected in: “The in vivo toxicity, immunogenicity, and  metabolic pathways of inorganic nanoparticles, especially for chiral self-assembled structures containing chromium and gold, will restrict their clinical applications”

References:

many of the name of the journals are not abbreviated

a lot of references reported: “doi:https://doi.org/” or “doi:doi:” instead of “doi.org/”

please correct

Ref 1 https://doi.org/10.1016/B978-0-444-64027-7.00001-Xpp. 3-28

Ref 52 l-amino acid. 2018, 4, eaas9819, doi:doi:10.1126/sciadv.aas9819

Ref 56: 10.1101/2020.12.28.424604 %J bioRxiv, 2020.2012.2028.424604, doi:10.1101/2020.12.28.424604 %J bioRxiv.

Ref 66 Mazur, M.K.A.P.A.H.-S.B.O.-M.B.G.W.B.A.f.t.P.o.E.H.D.f.B.; Evaluation of Their Cytotoxic, A. In Catalysts, 2020; Vol. 10.

Ref 74 Asmari, M.W.X.C.N.P.M.K.S.L.L.H.R.S.A.U.E.D.S.C.M.S.-B.H.C.f.E.S.; Determination: Functionalization of Chiral, S.; Recognition of Selector-Selectand, I. In Molecules, 2021; Vol. 26

Author Response

We would like to thank all the Reviewers for their time and effort invested in the evaluation of our manuscript. We thoroughly considered all the comments and extensive editorial changes were made in the main text. Please find below or point-by-point replies to the critique accompanied by brief descriptions of changes made in the manuscript.

Reviewer: 1

In this review intitled “Biomimetic self-assembled chiral inorganic structures: a new strategy for solving medical problems” authors report various forms of chiral self-assembled inorganic structures and applications in biomedical fields of biomimetic chiral self-assembled particles. It is a valuable source of information in this field. Only technical correction should be done:

  1. Line 31-32 “chiral organics” have to be corrected in “chiral organic compounds”

Line 92-93 “l- or d-“ is “L- or D-“

Line220 “H2O2” is “H2O2”

Line 225 and 226: “d-Co(OH)2 NPs”  have to be corrected in: “D-Co(OH)2 NPs”

Line 226:” l-Co(OH)2 NPs” have to be corrected in: “L- Co(OH)2 NPs”

Figure 1 and 2 are never cited in the text

Line 302-303: “The in vivo toxicity, immunogenicity, and  metabolic pathways of inorganic nanoparticles; especially for chiral self-assembled structures containing chromium and gold; will restrict their clinical applications” have to be corrected in: “The in vivo toxicity, immunogenicity, and  metabolic pathways of inorganic nanoparticles, especially for chiral self-assembled structures containing chromium and gold, will restrict their clinical applications”

Reply 1-1. We appreciate the suggestion for revision of these details and apologize for these techincal mistakes.All these problems have been revised in the revised manuscript.

  1. many of the name of the journals are not abbreviated

a lot of references reported: “doi:https://doi.org/” or “doi:doi:” instead of “doi.org/”

please correct

Ref 1 https://doi.org/10.1016/B978-0-444-64027-7.00001-Xpp. 3-28

Ref 52 l-amino acid. 2018, 4, eaas9819, doi:doi:10.1126/sciadv.aas9819

Ref 56: 10.1101/2020.12.28.424604 %J bioRxiv, 2020.2012.2028.424604, doi:10.1101/2020.12.28.424604 %J bioRxiv.

Ref 66 Mazur, M.K.A.P.A.H.-S.B.O.-M.B.G.W.B.A.f.t.P.o.E.H.D.f.B.; Evaluation of Their Cytotoxic, A. In Catalysts, 2020; Vol. 10.

Ref 74 Asmari, M.W.X.C.N.P.M.K.S.L.L.H.R.S.A.U.E.D.S.C.M.S.-B.H.C.f.E.S.; Determination: Functionalization of Chiral, S.; Recognition of Selector-Selectand, I. In Molecules, 2021; Vol. 26

Reply 1-2. Thank you very much for pointing these out. We checked all references and revised the name of journals and the format of their DOI numbers, and corrected the wrong ones.

Reviewer 2 Report

The manuscript “Biomimetic self-assembled chiral inorganic structures: a new strategy for solving medical problems” reviewed various methods to synthesize biomimetic self-assembled chiral structures, and summarized the medical applications of self-assembled chiral inorganic structures on (a) promoting cell adhesion, proliferation and differentiation, (b) functioning as molecular probes for chiral identification or as chiral fluorescent probes which might have higher thermal stability and photostability, fluorescence quantum yield and large Stokes shift, and (c) anti-tumor therapies via killing of tumor cells by inducing autophagy and apoptosis of tumor cells or photothermal effect of chiral self-assembled structures, or via causing immune-induced responses adverse to tumor cells. The manuscript is informative and well organized which will benefit peers to grasp a clear picture in the research area of self-assembled chiral inorganic structures. I would recommend its publication on Biomimetics with following minor revisions:

1.       In the abstract in lines 17-18, “Researches of cell behavior regulation, molecular probes, and tumor therapy is a novel approach to chiral inorganic self-assembled structures…” should be “Researches of cell behavior regulation, molecular probes, and tumor therapy are novel approaches to chiral inorganic self-assembled structures…”.

2.       For images (a)-(d) in Figure 1, each image should be introduced in detail in the main context. Otherwise it seems Figure 1 is not linked with the context and is not necessary to be in the manuscript.

3.       For images (a)-(c) in Figure 2, each image should be introduced in detail in the main context.

4.       In Figure 1 and Figure 2, each image should get permission for use in the manuscript, and the description “Reproduced from [x], YYY” should be revised to “Reproduced with copyright permission from [x], YYY”.

5.       In line 140, “Depending on the density of the NCs”, is NC representing nanocubes or nanocrystals? To avoid confusion, the authors can replace NC with nanocubes or nanocrystals.  

Author Response

We would like to thank all the Reviewers for their time and effort invested in the evaluation of our manuscript. We thoroughly considered all the comments and extensive editorial changes were made in the main text. Please find below or point-by-point replies to the critique accompanied by brief descriptions of changes made in the manuscript.

Reviewer: 2

The manuscript “Biomimetic self-assembled chiral inorganic structures: a new strategy for solving medical problems” reviewed various methods to synthesize biomimetic self-assembled chiral structures, and summarized the medical applications of self-assembled chiral inorganic structures on (a) promoting cell adhesion, proliferation and differentiation, (b) functioning as molecular probes for chiral identification or as chiral fluorescent probes which might have higher thermal stability and photostability, fluorescence quantum yield and large Stokes shift, and (c) anti-tumor therapies via killing of tumor cells by inducing autophagy and apoptosis of tumor cells or photothermal effect of chiral self-assembled structures, or via causing immune-induced responses adverse to tumor cells. The manuscript is informative and well organized which will benefit peers to grasp a clear picture in the research area of self-assembled chiral inorganic structures. I would recommend its publication on Biomimetics with following minor revisions:

  1. In the abstract in lines 17-18, “Researches of cell behavior regulation, molecular probes, and tumor therapy is a novel approach to chiral inorganic self-assembled structures…” should be “Researches of cell behavior regulation, molecular probes, and tumor therapy are novel approaches to chiral inorganic self-assembled structures…”.

Reply 2-1. Many thanks to you for the reminder, some grammartical mistakes occured when we wrote the manuscript ,we had corrected them.

  1. For images (a)-(d) in Figure 1, each image should be introduced in detail in the main context. Otherwise it seems Figure 1 is not linked with the context and is not necessary to be in the manuscript.
  2. For images (a)-(c) in Figure 2, each image should be introduced in detail in the main context.

Reply 2-2,2-3. We agree that our figure should be introduced in our main context. Figure 1a,1b,1c are the reported structures we cited to show multiple complex levels of chiral self-assembly structures, so we marked them in our main text. And we added details to introduced other figures in our main text.

  1. In Figure 1 and Figure 2, each image should get permission for use in the manuscript, and the description “Reproduced from [x], YYY” should be revised to “Reproduced with copyright permission from [x], YYY”.

Reply 2-4. Thanks a lot for pointing them out, we had corrected these problems. We attached our permission files with our manuscript.

  1. In line 140, “Depending on the density of the NCs”, is NC representing nanocubes or nanocrystals? To avoid confusion, the authors can replace NC with nanocubes or nanocrystals.

Reply 2-5. Thanks a lot for pointing it out, we checked the original article and confirmed that NCs means nanocrystals, and we had replaced it.

Reviewer 3 Report

This is a short review about self-assembled chiral inorganic structures, and their applications in medicine.

I am sorry to say that the idea could be good, but the result is not so good.

The text, despite being short is difficult to read, paragraphs are very long and not very readable, probably because of punctation. E.g.: L20-21, “It is worth considering whether there is some correspondence or juxtaposition between the chirality-induced selection effect exhibited by chiral self-assembled structures in the field of chiral photonics and the selectivity exhibited by chiral isomers in the medical field”. This sentence is very difficult without breathing.

In the introduction, references are to general.E.g.: L52 “The reported diversity of chiral self-assembly structures; rod, cube, propeller, helix, flower, fan, and hedgehog chiral particles[17-22].” References should be indicated for each case.

Figures should be named and referenced indicated throughout the text, rather than a block from time to time, in the middle of the text

Very importantly, I miss a lot of references to helical metal complexes interacting with DNA, among other aspects.

Finally, What is this reference

74. Asmari, M.W.X.C.N.P.M.K.S.L.L.H.R.S.A.U.E.D.S.C.M.S.-B.H.C.f.E.S.; Determination: Functionalization of Chiral, S.; Recognition of Selector-Selectand, I. In Molecules, 2021; Vol. 26.

Author Response

We would like to thank all the Reviewers for their time and effort invested in the evaluation of our manuscript. We thoroughly considered all the comments and extensive editorial changes were made in the main text. Please find below or point-by-point replies to the critique accompanied by brief descriptions of changes made in the manuscript.

Reviewer: 3

This is a short review about self-assembled chiral inorganic structures, and their applications in medicine.

I am sorry to say that the idea could be good, but the result is not so good.

  1. The text, despite being short is difficult to read, paragraphs are very long and not very readable, probably because of punctation. E.g.: L20-21, “It is worth considering whether there is some correspondence or juxtaposition between the chirality-induced selection effect exhibited by chiral self-assembled structures in the field of chiral photonics and the selectivity exhibited by chiral isomers in the medical field”. This sentence is very difficult without breathing.

Reply 3-1. Thank you very much for your suggestion. We tried to express our views in as concise sentences as possible and resulted in the brevity of the article. This long sentence has been revised for readability.

  1. In the introduction, references are to general.E.g.: L52 “The reported diversity of chiral self-assembly structures; rod, cube, propeller, helix, flower, fan, and hedgehog chiral particles[17-22].” References should be indicated for each case.

Reply 3-2. Thanks for reminding! Point-to-point references have been added in the revised manuscript.

  1. Figures should be named and referenced indicated throughout the text, rather than a block from time to time, in the middle of the text

Reply 3-3. Thank you for your suggestion. Missing image annotations have been filled in the full text of the revised manuscript.

  1. Very importantly, I miss a lot of references to helical metal complexes interacting with DNA, among other aspects.

Reply 3-4. We totally agree with your suggestion. Chiral assemblies of metals modified with DNA as ligands are indeed a very important aspect of this system. New content was added in the revised manuscript.

5.Finally, What is this reference

  1. Asmari, M.W.X.C.N.P.M.K.S.L.L.H.R.S.A.U.E.D.S.C.M.S.-B.H.C.f.E.S.; Determination: Functionalization of Chiral, S.; Recognition of Selector-Selectand, I. In Molecules, 2021; Vol. 26.

Reply 3-5. We apologize for the wrong citation format. A disorder in the automatic citation software led to this undesirable result. We have rechecked the formatting of all citations including this one in the revised manuscript.

Reviewer 4 Report

The article is devoted to the study of chiral inorganic structures, as well as the prospects for expanding the functional boundaries of inorganic materials. In general, the presented study is quite interesting and promising not only from a fundamental point of view, but also from a further practical application. The article corresponds to the declared journal and can be accepted for publication in the future after the authors answer a number of questions that have arisen during its analysis.

1. The abstract needs to be improved, the authors should reflect in more detail the novelty and practical significance of the work.

2. The authors should give a more detailed description of the prospects for using these structures in biomedicine in the introduction.

3. The presented graphs and images in the figures require a significant improvement in quality, since in most cases the details are hardly distinguishable on them.

Author Response

We would like to thank all the Reviewers for their time and effort invested in the evaluation of our manuscript. We thoroughly considered all the comments and extensive editorial changes were made in the main text. Please find below or point-by-point replies to the critique accompanied by brief descriptions of changes made in the manuscript.

Reviewer: 4

The article is devoted to the study of chiral inorganic structures, as well as the prospects for expanding the functional boundaries of inorganic materials. In general, the presented study is quite interesting and promising not only from a fundamental point of view, but also from a further practical application. The article corresponds to the declared journal and can be accepted for publication in the future after the authors answer a number of questions that have arisen during its analysis.

  1. The abstract needs to be improved, the authors should reflect in more detail the novelty and practical significance of the work.

Reply 4-1. Thanks for your valuable comments. A detailed description is added in the abstract of the revised manuscript.

  1. The authors should give a more detailed description of the prospects for using these structures in biomedicine in the introduction.

Reply 4-2. We completely agree with what you have pointed out. Some detailed descriptions have been added in the revised manuscript.

  1. The presented graphs and images in the figures require a significant improvement in quality, since in most cases the details are hardly distinguishable on them.

Reply 4-3. Thanks a lot for your advice. We had changed our graphs to improve them in quality, however we reproduced them from their original writers so we afraid that we can’t provide perfect graphs or images.

Round 2

Reviewer 3 Report

I think that this is a short review that only deals with nanomaterials, not with inorganic structures, as it does not consider any coordination compound. Therefore, instead of ”Biomimetic self-assembled chiral inorganic structures: a new strategy for solving medical problems”, it should be entitled: “Biomimetic self-assembled chiral inorganic nanomaterials: a new strategy for solving medical problems”

Since metal complexes are not considered at all, after this change, this mini-review could be considered for publication after taking into account the following considerations:

At this moment, the only mention to AuNRs interacting with DNA is: Researchers have found that self-assembly can be controled through integration of the gold nanoparticles[39]. Similarly, Kuzyk et al. used DNA origami technology to synthesize gold nanoparticle helical[40]. By designing the ‘X’ pattern of the arrangement of DNA capturing strands (15nt) on both sides of a two-dimensional DNA origami template, AuNRs functionalized with the complementary DNA sequences were positioned on the origami and were assembled into AuNR helices with the origami intercalated between neighboring AuNRs [37]. (Figure 1d)[37] . If authors consider that this is a thorough revision, at least they should take into account some serious previous review as https://doi.org/10.1021/acs.chemrev.1c00422.

By the way, L110 AuNRs [37 ] this is mentioned before the figure 1, so nanorods should be mentioned.

Author Response

Reviewer: 3,Round 2

I think that this is a short review that only deals with nanomaterials, not with inorganic structures, as it does not consider any coordination compound. Therefore, instead of ”Biomimetic self-assembled chiral inorganic structures: a new strategy for solving medical problems”, it should be entitled: “Biomimetic self-assembled chiral inorganic nanomaterials: a new strategy for solving medical problems”

Reply 3(2)-1. Thank you very much for your suggestion. “Structures” in our title is truly a large range and the main text is not sufficient enough. After discussion, we decide to change the title according your advice.

Since metal complexes are not considered at all, after this change, this mini-review could be considered for publication after taking into account the following considerations:

At this moment, the only mention to AuNRs interacting with DNA is: Researchers have found that self-assembly can be controled through integration of the gold nanoparticles[39]. Similarly, Kuzyk et al. used DNA origami technology to synthesize gold nanoparticle helical[40]. By designing the ‘X’ pattern of the arrangement of DNA capturing strands (15nt) on both sides of a two-dimensional DNA origami template, AuNRs functionalized with the complementary DNA sequences were positioned on the origami and were assembled into AuNR helices with the origami intercalated between neighboring AuNRs [37]. (Figure 1d)[37] . If authors consider that this is a thorough revision, at least they should take into account some serious previous review as https://doi.org/10.1021/acs.chemrev.1c00422.

Reply 3(2)-2. We agree that AuNRs are very advanced nanomaterials with great potential in the chiral self-assembly area. So we added contents about the DNA bound nanoparticles, including AuNPs and other inorganic nanostructure, and involved other structure, such as pyramid. Besides, hierarchical assembly is also mentioned.

By the way, L110 AuNRs [37 ] this is mentioned before the figure 1, so nanorods should be mentioned.

Reply 3(2)-3. Thank you for your advice. We have added more examples about the nanorods, not only the gold nanorods, but also other advanced and cutting-edge nanomaterials. And we enumerated nanorods with other important structures in the introduction. 
